# Study of Microbial Cultures for the Bioleaching of Scandium from Alumina Industry By-Products [†]

Kyriaki Kiskira [1,*], Theopisti Lymperopoulou [2], Lamprini-Areti Tsakanika [1], Charalampos Pavlopoulos [3], Konstantina Papadopoulou [3], Klaus-Michael Ochsenkühn [1], Gerasimos Lyberatos [3] and Maria Ochsenkühn-Petropoulou [1]

[1]  Laboratory of Inorganic and Analytical Chemistry, School of Chemical Engineering, Zografou Campus, National Technical University of Athens, Iroon Polytechniou 9, 15773 Athens, Greece; betsakan@central.ntua.gr (L.-A.T.); klausochsenkuehn@yahoo.com (K.-M.O.); oxenki@central.ntua.gr (M.O.-P.)
[2]  Products and Operations Quality Control Laboratory School of Chemical Engineering, School of Chemical Engineering, Zografou Campus, National Technical University of Athens, Iroon Polytechniou 9, 15773 Athens, Greece; veralyb@chemeng.ntua.gr
[3]  Laboratory of Organic Chemical Technology, School of Chemical Engineering, Zografou Campus, National Technical University of Athens, Iroon Polytechniou 9, 15773 Athens, Greece; charpavlop@gmail.com (C.P.); kpapado@chemeng.ntua.gr (K.P.); lyberatos@chemeng.ntua.gr (G.L.)
*  Correspondence: kirki.kis@gmail.com; Tel.: +0030-698-5808-614
†  This paper is part of the oral presentation entitled "Use of Different Microbial Cultures for Bioleaching of Scandium from Bauxite Residue" presented at ERES2020, 3rd Conference on European Rare Earth Resources, 6–9 October 2020, virtual, https://eres2020.eres-conference.eu/.

**Abstract:** The disposal of voluminous, highly alkaline, bauxite residue (BR), the industrial by-product of alumina production by the Bayer process, constitutes an intricate global environmental problem. BR, containing valuable metals such as rare-earth elements (REEs)—in particular, scandium (Sc)—can be used as a secondary source for REE extraction. The scope of this study was the investigation of bioleaching as an innovative and environmentally friendly approach for the extraction of Sc from BR. The bioleaching parameters were studied on Greek BR and experiments were performed using different microbial cultures and solid to liquid ratios (S/L). The maximum extraction of Sc was 42% using *Acetobacter tropicalis* in a one-step bioleaching process at 1% S/L. The main organic acids produced were acetic, oxalic, and citric. The bioleaching data indicated a probable synergistic effect of the different organic acids produced by microorganisms along with a more targeted leaching mechanism.

**Keywords:** bauxite residue; scandium; bioleaching; acetobacter; industrial by-product; rare earth elements; red mud

## 1. Introduction

Bauxite residue (BR), also called red mud, is a highly alkaline waste by-product generated from the Bayer process for alumina production. The disposal of BR constitutes a global environmental concern, as it is considered a hazardous material due to its alkalinity and volume. A huge quantity equaling 120 million tons (Mtpa) is generated annually and more than 3.5 billion tons have already been stockpiled in storage areas [1].

BR consists mainly of ferrous minerals, aluminum oxides, silicon, titanium, calcium, and sodium compounds [2]. Its composition varies and depends on the bauxite origin and the Bayer and sintering process parameters. As far as the rare earth elements (REEs) concentrations are concerned, it has been found that the Greek BR is enriched by a factor of 2 compared with the original bauxite, reaching 1 kg/ton. The high scandium (Sc) content ~0.1 kg/ton, almost constant during the last 25 years, is close to that found in the Sc main resources and represents 90–92% of the economic value for all REEs [2,3]. Many investigations have been performed on the utilization of BR for numerous purposes, aiming

to reduce stockpiling and introduce it in the economic cycle. In the last few decades, the increasing use of Sc and other REEs in high tech, aerospace, and defense applications have resulted in a higher demand for these elements and their classification as critical raw materials (CRMs) by the European Commission due to their supply risk [4]. The utilization of BR as a secondary source, especially for Sc and REEs, has been studied by several researchers [5–7]. Hydrometallurgical methods were mainly investigated using different mineral acids, such as nitric [3], hydrochloric [7], and sulfuric acid [2], under different leaching conditions [7,8].

Ochsenkühn-Petropoulou et al. developed an integrated viable method for the recovery of Sc and REEs, which includes hydrometallurgical treatment by dilute nitric acid (leaching), ion exchange separation, a selective extraction/back-stripping process, and chromatographic techniques for the individual separation of elements at high purity [9,10]. The method was partially scaled up in the pilot plant for Sc selective purification. A high Sc recovery of about 75% was achieved, while other REEs' recovery ranged between 30 and 60% [9,11]. The final product consisted of 99% Sc and 1% Al [11]. However, since the hydrometallurgical treatment with $HNO_3$ has a negative environmental impact, the use of sulfuric acid ($H_2SO_4$) as an alternative leaching agent was also investigated. Although environmentally more friendly and less expensive than $HNO_3$, $H_2SO_4$ led to a lower yield of REEs and up to a 55% recovery of Sc [12–14].

Ionic liquids represent an alternative approach for Sc and REE recovery [15,16]. Davris et al. [15] reported high recoveries of REEs (70–85%); however, the process is expensive due to the high price of ionic liquids and requires further investigation for its use on an industrial scale. A combination of pyro- and hydrometallurgical processes for recovering iron and the leaching of Sc has also been reported [7,17]. Despite the high recoveries, pyrometallurgical processes require considerable energy inputs. Furthermore, there is a lack of information on cost-effective estimation in real applications in all of these processes [2,14]. Concerning organic acids, recovery yields of up to 50% were obtained for all REEs [7]. However, abiotic leaching with laboratory-prepared solutions is not as effective as bioleaching, which results in higher Sc recovery percentages according to research conducted in this field [18].

Biotechnologies based on microorganisms and their interaction with different materials can play an important role in metal recovery [19,20]. Bioleaching is a technology considered as 'a green technology', with operational flexibility and low energy requirements [18]. Bioleaching can be performed by chemoautotrophic and chemoheterotrophic microorganisms; however, due to the highly alkaline environment of BR, chemoheterotrophics are more suitable for BR, as they have a higher tolerance to heavy metals and a faster growth and adaptation in a wide range of pHs [21,22]. Furthermore, chemoheterotrophic microorganisms can use metabolites, such as proteins and amino acids, in BR to form complexes with toxic metal ions [23].

Many studies have investigated the bioleaching of different ores and industrial wastes [24–26]; only a few, however, involve the bioleaching of BR, especially for REE extraction [18,22,27]. Fungal strains such as *Penicillium tricolor* [18,27] and *Aspergillus niger* [27–29] and microalgae [30] such as *Desmodesmus quadricauda*, *Chlamydomonas reinhardtii*, and *Parachlorella kessleri* have been investigated. However, the main disadvantages of the use of fungi and microalgae compared to bacteria are the generation of a large amount of biomass and spores, which increase the cost of the process and cause secondary pollution, and the separation of BR particles from fungal biomass after bioleaching is challenging [22]. Thus, Qu et al. [22] investigated the bioleaching process with the use of chemoheterotrophic *Acetobacter* sp. bacterium isolated from BR, which resulted in a 52% Sc recovery. However, further investigation is needed regarding the ability of other microorganisms—such as *Acetobacter tropicalis*, a commercially available microorganism—and mixed cultures to produce organic acids involved in the bioleaching of BR.

The objectives of this work were (1) to investigate the development of a bioleaching procedure for Sc recovery from Greek BR due to its high content of Sc compared to other

REEs and the optimization of the process by testing the effect of different microorganisms, BR (S/L) ratio, biomass production, final pH, and percentage recovery of Sc in batch bioassays and (2) to determine the organic acids produced by the microorganisms.

## 2. Materials and Methods

### 2.1. Bauxite Residue

The BR was provided by Mytilineos S.A., Metallurgy Unit, which is located at Agios Nikolaos, Viotia, Greece. All experiments were conducted using BR (batch 2014). Characterization of BR was performed by XRD (X-ray Diffraction, D8 ADVANCED TWIN–TWIN XRD spectrometer with detector LYNEXEYE (1D Mode), BRUKER, Karlsruhe, Germany) and SEM (Scanning Electron Microscopy, FEI Quanta 200, Hillsboro, USA) equipped with EDAX for semiquantitative chemical analysis (as shown in Figures S1 and S2, and Table S1 in the Supplementary Material). The BR (red mud) had a pH value of 11.3 and a moisture content of around 25%. The BR was dried at 105 °C for 24 h prior to the bioleaching experiments.

### 2.2. Sources of Microorganisms and Cultivation Mineral Media

The cultures used in this study were: (a) digestate (anaerobic digestion effluent) collected from a pilot-scale anaerobic digester and (b) a chemoheterotrophic Bacterium, *Acetobacter tropicalis,* which was purchased from the 'Leibniz-Institute DSMZ-German collection of microorganisms and cell cultures' in Braunschweig (Germany).

The *Acetobacter tropicalis* culture was cultivated in a mineral medium of the following composition ($g \cdot L^{-1}$): 5.0 bacto peptone, 5.0 yeast extract, 5.0 glucose, and 1.0 $MgSO_4 \cdot 7H_2O$. The medium was sterilized by autoclaving at 121 °C for 15 min. The trace mineral solution was added from a sterile stock solution and was prepared by dissolving the following compounds in a 1.5 $g \cdot L^{-1}$ nitrilotriacetic acid disodium salt solution (quantities were reported in $g \cdot L^{-1}$): 0.50 $MnSO_4$, 1.00 NaCl, 0.10 $FeSO_4 \cdot 7H_2O$, 0.10 $CaCl_2 \cdot 2H_2O$, 0.10 $CoCl_2 \cdot 6H_2O$, 0.13 ZnCl, 0.01 $CuSO_4 \cdot 5H_2O$, 0.01 $AlK(SO_4)_2 \cdot 12H_2O$, 0.01 $H_3BO_3$, and 0.025 $Na_2MoO_4 \cdot 2H_2O$ [31,32].

Digestate (anaerobic digestion (AD) effluent) was collected from a pilot-scale continuous stirred tank (CSTR) methanogenic reactor with an operating volume of $4m^3$, operated under mesophilic conditions (35–40 °C) with a hydraulic retention time (HRT) of 20 d. More specifically, this pilot scale unit focused on the effectiveness and benefits of using pretreated food waste (dried and shredded) as a feedstock for methane ($CH_4$) production via anaerobic digestion [33].

### 2.3. Bioleaching Experiments and Sampling

BMP tests were conducted with the Automatic Methane Potential Test System II (AMPTS II; Bioprocess Control, Sweden) cultivated with digestate. The operating volume was 450 mL, and all tests were carried out under mesophilic conditions (35 ± 1 °C), at least until the daily gas production was <1% of the cumulative total gas production for 3 consecutive days. Prior to incubation, all bottles were flushed with $N_2$. The AMPTS II system reports the cumulative methane volume from each bottle, which can be used to calculate the final BMP. A BR of 10, 20, and 30% S/L- pulp density was added to the bottles.

*Acetobacter tropicalis* culture was inoculated into a leaching medium and incubated at 30 °C and 120 rpm in an orbital shaking incubator for 1 month for activation.

Bioleaching was performed using a 10 and a 20% $w/w$ of bacterium suspension inoculated in 250 mL Erlenmeyer flasks with 150 mL of a leaching medium with different S/L ratios of sterilized BR. Experiments were conducted at 20 °C (room temperature) and at 30 °C (incubator).

The pre-treatment of BR involved a pH adjustment to 7, with 0.1 M HCl in all of the experiments. The bioleaching period was 30 d, and the bioleaching mode was a one-step process; the bacterium was inoculated in the leaching medium together with BR. Sampling was performed by taking 5–10 mL samples at regular intervals to analyze the biomass concentration, pH value, metal ions, and organic acid concentration. All the culture media

involved in this study were sterilized by autoclaving at 121 °C for 15 min before use. All batch tests were conducted in triplicate, and control samples were tested.

The Sc extraction degree was calculated based on the following Equations (1) and (2):

$$C_{(Sc\ extracted\ in\ \mu g/g)} = (C_{(Sc\ extracted\ in\ \mu g/mL)} * V_{(leaching\ solution\ in\ mL)})/m_{(mass\ of\ dry\ BR\ leached\ in\ g)}, \tag{1}$$

$$\%\ Sc\ recovery = (C_{(Sc\ extracted\ in\ \mu g/g)}/C_{(Sc\ in\ BR\ in\ \mu g/g)}) * 100. \tag{2}$$

### 2.4. Analytical Methods

A chemical analysis for Sc (361.383 nm) and Fe (248.3 nm) of the leachate solutions after filtration at 0.45 μm under vacuum was conducted in triplicate after the appropriate dilution using ICP-OES Optima 7000DV supplied by PerkinElmer (Waltham, MA, USA); additionally, for Fe, AAS AA240FS (acetylene/air, lamp current 5 mA) supplied by Varian (Palo Alto, CA, USA), respectively, was used. The measurements of TSS (Total Suspended Solids) and VSS (Volatile Suspended Solids) for biomass determination were carried out according to Standard Methods [34]. The identification of VFAs (Volatile Fatty Acids) and organic acids was performed with a Shimadzu (Kyoto, Japan) (GC2010) gas chromatograph and an Agilent Technologies (Santa Clara, USA)1260 Infinity II HPLC, with a column Agilent Hi-plex H of 300 mm × 7.7 mm. The operating condition of HPLC: Eluent: Sulfuric Acid 5 mM (Aquatic), Flow Rate: 0.4 mL/min, Time: 75 min, Column Temperature: 50 °C, Injection Volume: 20 μL, and Detector: DAD (UV at 210 nm). The pH was measured using an X digital pH-meter. The number of bacterial cells during bacterial growth of *Acetobacter tropicalis* was counted by a standard plate count (SPC) method. Additionally, the determination of the bacterial growth of *Acetobacter Tropicalis* was performed by measuring the optical density at 600 nm with an X spectrophotometer. XRD analysis of BR before and after bioleaching was performed using D8 ADVANCED TWIN–TWIN XRD spectrometer with detector LYNEXEYE (1D Mode) supplied by BRUKER (Karlsruhe, Germany). The micro-morphology of the samples after sputter coating with gold before and after bioleaching was also examined by a scanning electron microscope (SEM, FEI Quanta 200, 20 keV, solid state detector-SSD, Hillsboro, USA) equipped with EDAX for semi-quantitative chemical analysis.

## 3. Results and Discussion

The initial Sc concentration of the BR 2014 batch used in this study was determined by ICP-OES at 104.7 ± 3.0 μg of Sc/g of BR after a lithium tetraborate fusion [2]. XRD diagrams and SEM images before and after bioleaching with their interpretation are attached as supplementary material in Figures S1 and S2. Table S1 shows the EDX data for the chemical composition (mass %) present in BR before and after bioleaching.

### 3.1. Preliminary Tests

Three different sets of experiments were conducted in order to determine the critical parameters. The first set was carried out using the digestate inoculum at 35 °C, while in the other two a 10% *w/w* of *Acetobacter tropicalis* suspension was used. The second set was incubated at 30 °C and the third one at 20 °C. The incubation time was 30 days, and the response factors were the pH and the percentage recovery of Sc. All experiments were implemented in 10, 20, and 30% pulp density solutions (S/L) of sterilized BR. Results are presented in Tables 1 and 2.

**Table 1.** Sc recovery over time for different culture solutions.

| Culture | *Acetobacter tropicalis* | | | | | | Digestate Inoculum | | |
|---|---|---|---|---|---|---|---|---|---|
| Temp | 30 °C | | | 20 °C | | | 30 °C | | |
| S/L | 10% | 20% | 30% | 10% | 20% | 30% | 10% | 20% | 30% |
| days | % Sc recovery | | | | | | | | |
| 1 | 20.4 | 5.5 | 2.3 | 8.2 | 1.9 | 0.6 | 9.9 | 3.7 | 2.9 |
| 5 | 24.7 | 10.6 | 2.3 | 9.0 | 2.1 | 0.6 | 18.1 | 8.4 | 4.8 |
| 10 | 34.6 | 11.5 | 2.3 | 24.2 | 1.7 | 0.5 | 29.9 | 11.1 | 5.2 |
| 30 | 38.4 | 14.4 | 2.2 | 25.1 | 1.9 | 0.5 | 30.0 | 12.1 | 5.2 |

**Table 2.** Time-monitored pH values of different culture solutions.

| Culture | *Acetobacter tropicalis* | | | | | | Digestate Inoculum | | |
|---|---|---|---|---|---|---|---|---|---|
| Temp | 30 °C | | | 20 °C | | | 30 °C | | |
| S/L | 10% | 20% | 30% | 10% | 20% | 30% | 10% | 20% | 30% |
| days | pH | | | | | | | | |
| 1 | 3.2 | 3.6 | 3.4 | 3.5 | 4.5 | 4.9 | 5.1 | 6.8 | 7.0 |
| 5 | 2.4 | 3.0 | 3.1 | 3.6 | 4.3 | 4.8 | 3.6 | 4.9 | 6.9 |
| 10 | 2.0 | 3.0 | 3.1 | 3.0 | 4.2 | 4.8 | 3.0 | 4.2 | 5.6 |
| 30 | 1.9 | 2.9 | 3.1 | 2.6 | 4.2 | 4.8 | 2.7 | 4.0 | 5.4 |

Preliminary experiments were monitored for 30 days and, as can be seen, a plateau was reached after the 10th day in almost all cases. Initial tests involved the use of a digestate inoculum collected from a pilot-scale anaerobic digester in order to investigate the possibility of utilizing a less-costly crude product instead of a pure microorganism. As presented in Table 1, the inoculum seemed to act more slowly and resulted in less recovery. The S/L ratio was proven to be of critical importance under all conditions, ending in a negligible recovery for the 30% S/L. Experiments at room temperature also showed a lower Sc recovery, as expected. The effect of temperature was tested in order to obtain an estimate of the necessity of a heating unit in case of large-scale application. A similar trend is also reflected at pH value, which dropped down to 2 for the *Acetobacter* culture in the 10% S/L solution at 30 °C, while it seemed practically unaffected in the case of the sludge inoculum in the 30% S/L solution.

### 3.2. Parameters Evaluation

Bioleaching experiments under a one-step process were performed at 30 °C for 20 days using a 10 and 20% *w/w* of *Acetobacter tropicalis* suspension. The S/L ratios of BR solutions employed were 1, 2, 10, and 20%. The response factors were biomass production, final pH, and percentage recovery of Sc. All results are presented in Figure 1.



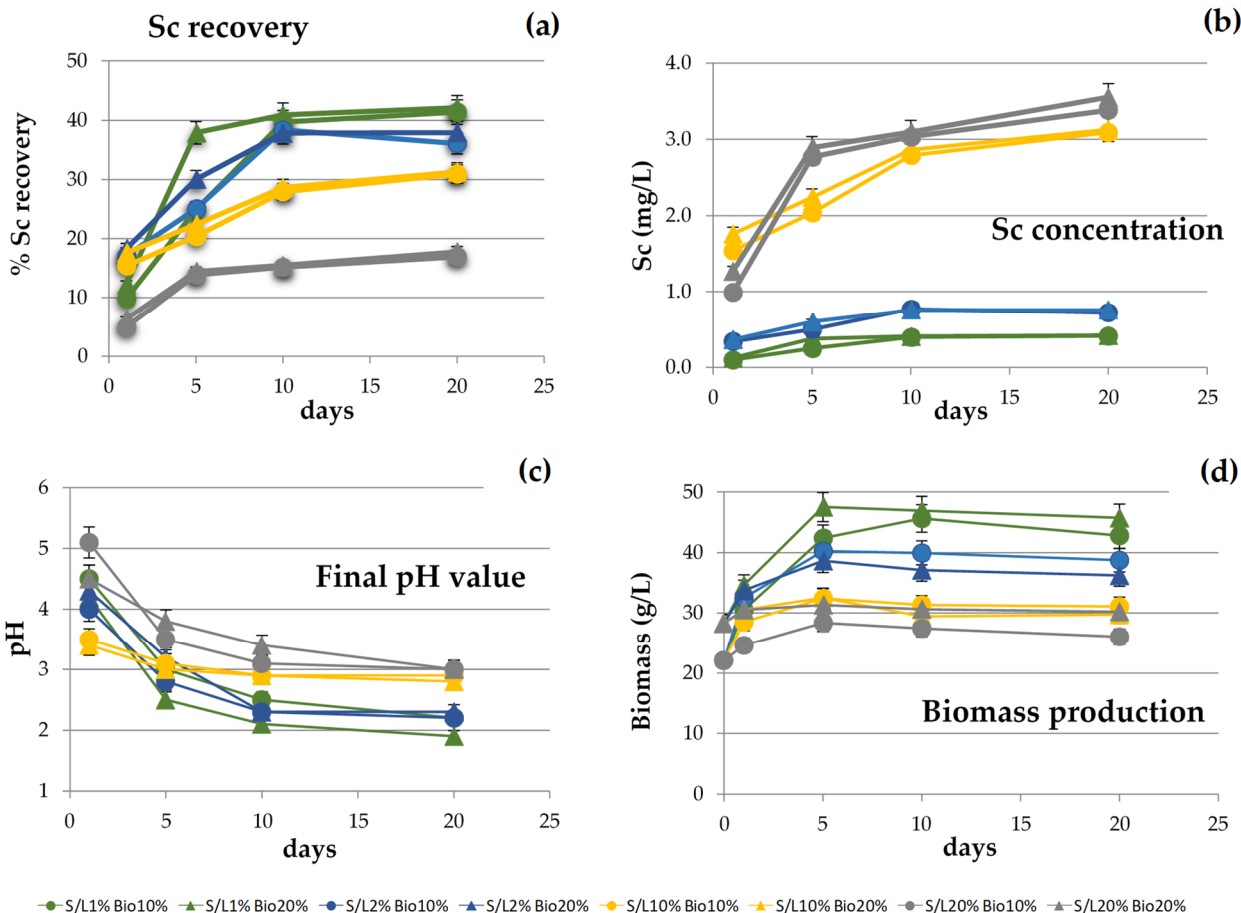

**Figure 1.** (**a**) Sc recovery, (**b**) Sc concentration, (**c**) pH of solution, and (**d**) biomass production by *Acetobacter tropicalis* in bioleaching experiments of 1, 2, 10, and 20% S/L- pulp density of BR with 10 and 20% *w/w* of bacterium suspension versus time.

Taking into consideration the preliminary tests, the effect of the S/L ratio, as well as the biomass dosage, were thoroughly investigated and monitored for up to 20 days, since a longer duration had no significant effect. The solutions of 1 and 2% S/L ratios exhibited the best Sc percentage recovery, increasing up to 42 and 38%, respectively, in agreement with other studies that reported higher recoveries at low S/L ratios [22,28]. Qu et al. [22] reported a higher recovery of Sc of 52% under a one-step process at a 2% S/L ratio using an *Acetobacter* sp. isolated from RM; this is, therefore, probably a more suitable microorganism for the bioleaching of BR.

When the biomass load was doubled, there was a slight increase in the Sc recovery, but this is of little practical interest in respect to the final recovery between the 10th and 20th day. The biomass increase influences the rate of recovery increase, especially at low S/L ratios, resulting in a considerable recovery already on the 5th day when a 1% S/L ratio is utilized (Figure 1a). A quite remarkable finding, related to Sc concentration, is that despite the lower recoveries, the solutions of 10 and 20% S/L ratios led to the highest Sc content (Figure 1b). Considering that the 20% pulp reached the plateau by the 5th day, the reduction in the bioleaching time seems quite realistic. The pH values kept on dropping until the 10th day in most cases, pointing out the relationship between acid production and Sc recovery (Figure 1c). Biomass production reached its highest value in the 1% S/L solution with the double biomass dosage, demonstrating the exact same trend as the Sc percentage recovery (Figure 1d).

The bioleaching period is quite long compared to acid leaching [14]. Nevertheless, biotechnologies are more widely used nowadays for their low cost and energy requirements [19]. Bioleaching is a self-sustaining technology since the microbial population

increases when the nutrients are present and decreases when the nutrients are limited [20]. However, the results obtained in this study are encouraging, especially in terms of the long-term process stability demonstrated by the *Acetobacter tropicalis*. The coupling of such cultures to the easy-to-operate showed an appealing resilience toward varying process conditions, which stimulate further research efforts in the field of bioleaching BR. Future developments should aim at enabling the process at an even lower bioleaching period and for the potential of the process to be compatible with the BR production for a possible future application in full-scale plants.

The recovery of iron in the final solution was also tested in order to estimate the selectivity of the method. The results are presented in Figure 2. The recovery of iron is consistently relatively low, rendering the bioleaching process quite selective, especially in the case of the 10% S/L solution and the 10% biomass dosage. This is in agreement with a previous bioleaching study that used *Acetobacter* sp. [22].

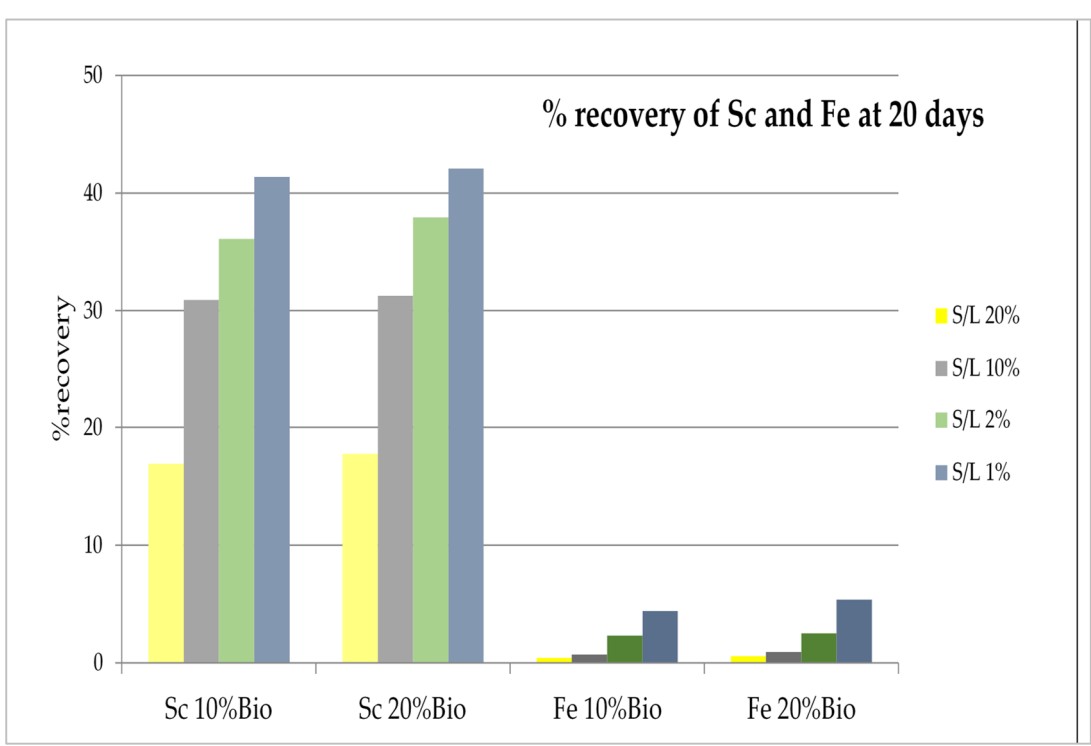

**Figure 2.** Sc and Fe recovery by *Acetobacter tropicalis* in bioleaching experiments of 1, 2, 10, and 20% S/L- pulp density of BR with 10 and 20% *w/w* of bacterium suspension on the 20th day.

### 3.3. Organic Acids Production

The investigation of organic acids produced by the bioleaching microorganisms is a key factor for the understanding of the bioleaching process [22]. The bioleaching efficiency can be determined by the organic acid production as the combination of oxygen and protons with water results in acidolysis, the detachment of metal ions from the surface of the leaching material [35]. Furthermore, organic acids chelate the metal ions in the solution and promote metal dissolution and complexation, reducing the metal toxicity [36].

Therefore, in this study, the determination of organic acids produced by *Acetobacter tropicalis* in bioleaching experiments was performed. Figure 3 shows the organic acid production in the bioleaching experiments at 1, 2, 10, and 20% S/L- pulp density of BR with 10 and 20% *w/w* of bacterium suspension in 20 d.

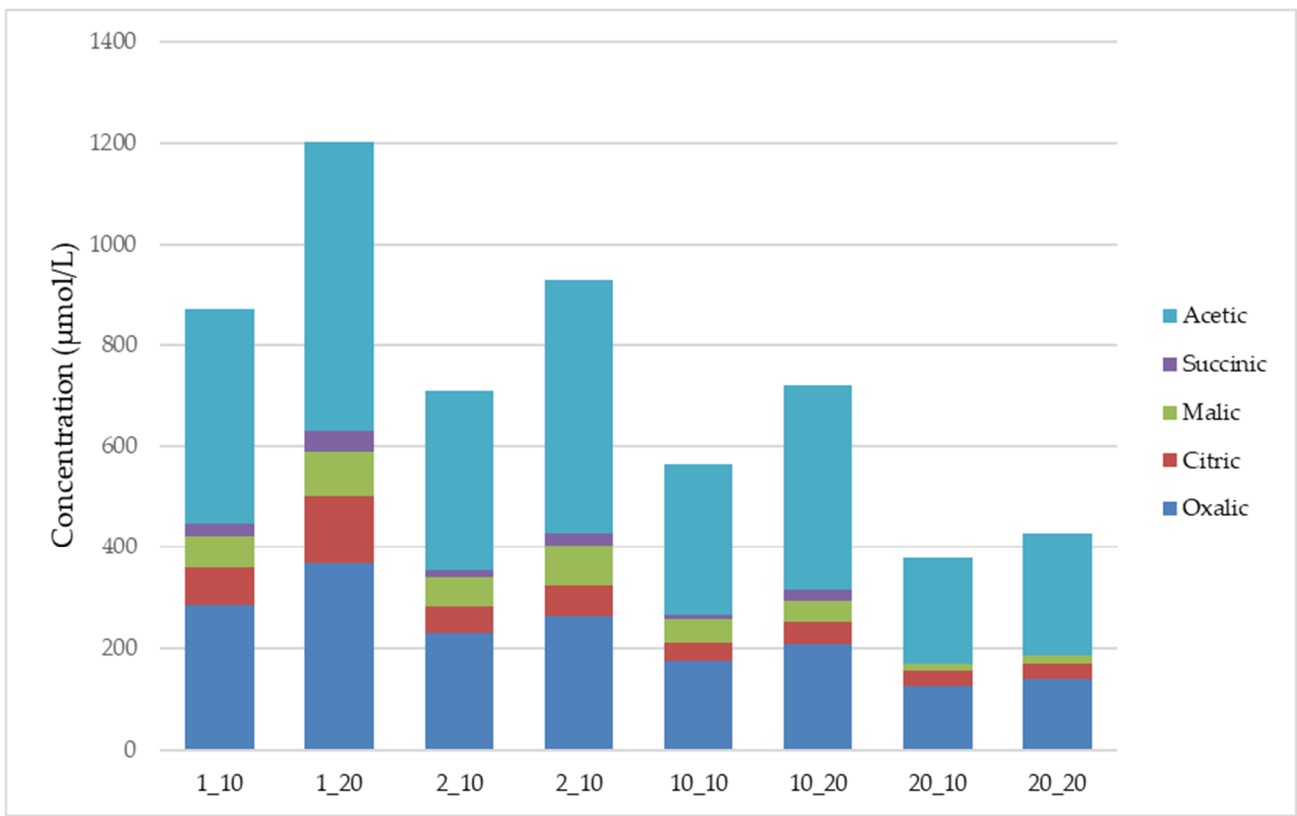

**Figure 3.** Organic acid production by *Acetobacter tropicalis* in bioleaching experiments of 1, 2, 10, and 20% S/L- pulp density of BR with 10 and 20% *w/w* of bacterium suspension in 20d (e.g., exp., 1_10: 1% S/L-pulp density of BR, 10% *w/w* bacterium suspension).

The bioleaching experiments with *Acetobacter tropicalis* with different parameters resulted mainly in acetic and oxalic acids, with lower concentrations of citric acid and minor quantities of maleic and succinic acid, except in the experiments with 20% BR pulp density where no succinic acid was detected. The different organic acids produced by *Acetobacter* agree with the results of other bioleaching studies [22]. The total organic acid production in all experiments decreased with the increase in pulp density. This is also in agreement with other bioleaching studies, stating that the citric acid concentration decreases with increasing pulp density when the bioleaching material contains Fe and Mn [36,37]. The oxalic acid was found in high concentrations, most probably due to the high pH values that promote the production of oxalic acid [38]. Oxalic acid with the REEs can form oxalates, resulting in an important detoxification mechanism that enhances microorganism growth [36].

The potential for the recirculation of the organic acids produced by microorganisms to increase their concentration is an important aspect. Further investigation is needed in order to evaluate the particular organic acids and decide on the "reuse strategy" and their ability to be effectively recycled and reused without losing their properties.

On the other hand, Qu et al. [22] reported that the total organic acid production increased with the increase in BR pulp density. These bioleaching experiments were performed, however, using an *Acetobacter* sp. isolated from a BR sample, and no pre-treatment of BR was involved (pH adjustment). When the initial pH of BR is around 11, microorganisms tend to retain the same conditions in their environment [21], by a higher production of acids as the alkalinity increased due to the increase in BR amount. The higher initial alkalinity may as well be the reason for the organic acid production of 150 mmol/L, which is higher than the 1200 µmol/L of this study (Figure 3). The higher amount of bacterium suspension resulted in an increase in organic acid production; however, in higher BR pulp densities, the increase was lower. The trend of organic acid

production is similar with the Sc recoveries, demonstrating that an indirect bioleaching with organic acid production was the main leaching mechanism.

A solution of 1% pulp density of BR with 20% *w/w* bacterium suspension resulted in the highest organic acid production, with acetic acid equal to 572.5 µmol/L being the main acid, as was expected from an *Acetobacter* strain, followed by oxalic (367.4 µmol/L), citric (134.3 µmol/L), malic (85.9 µmol/L), and succinic (42.0 µmol/L) (Figure 4). This agrees with the highest recovery of Sc that was recorded with the same conditions. The increase in organic acid concentration in experiments with 1% pulp density during the period of bioleaching is shown in Figure 4. The total organic acid concentration from 1 d to 20 d was increased three times.

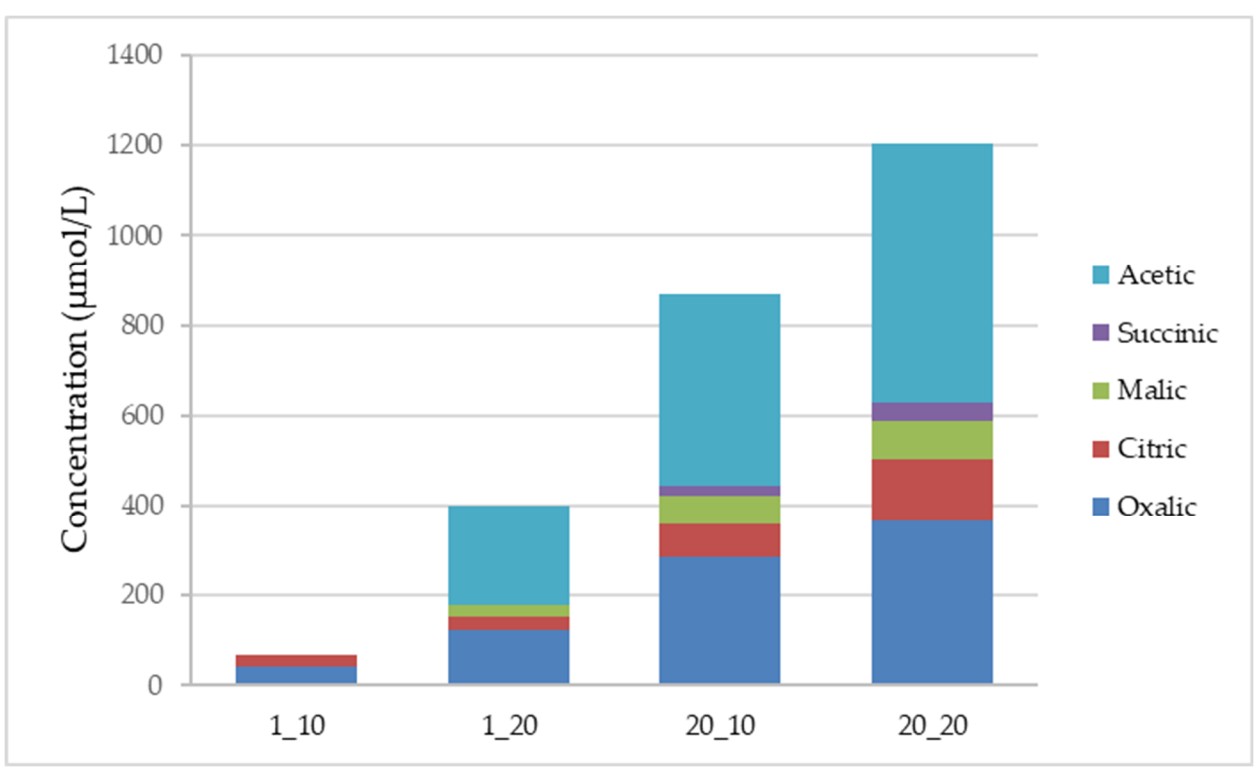

**Figure 4.** Organic acid production by *Acetobacter tropicalis* in bioleaching experiments of 1% S/L- BR pulp density with 10 and 20% *w/w* of bacterium suspension in 1 and 20 d. (e.g., exp., 1_10: 1% S/L-pulp density of BR, 10% *w/w* bacterium suspension).

## 4. Conclusions

This study provides a proof of concept for the bioleaching process of BR for the recovery of Sc. Preliminary experiments with different microorganisms showed that the pure culture of *Acetobacter tropicalis* resulted in the best bioleaching efficiency and it was chosen for further investigation. The highest Sc recovery of 42% was observed in the leachates with a 1% S/L- BR pulp density with 20% *w/w* of bacterium suspension, recorded after 20 days. Higher pulp densities were proven to reduce the Sc recovery. Bioleaching experiments with *Acetobacter tropicalis* resulted in mainly acetic and oxalic acids with lower concentrations of citric, malic, and succinic acids. The total organic acid production decreased with the increase in pulp density. The results suggest a synergistic effect of the different organic acids produced by microorganisms. Future developments should focus on lowering the incubation time and maximizing the Sc recovery in order to pave the way for the up-scaling of this promising REE extraction bioprocess.

**Supplementary Materials:** The following are available online at https://www.mdpi.com/article/10.3390/met11060951/s1: Figure S1: Mineralogical analysis of bauxite residue (BR) before and after bioleaching process with Acetobacter tropicalis. Figure S2: SEM images: (a) SEM image of BR before

bioleaching (1200×); (b) SEM image of BR before bioleaching (2500×); (c) SEM image of BR sample after 20d bioleaching using Acetobacter tropicalis in 20% *w/w* suspension with 1% S/L ratio of BR solution (2500×); (d) Bacterial cells-minerals and BR aggregates (Acetobacter tropicalis in 20% *w/w* suspension with 1% S/L ratio of BR solution) (5000×); (e) Cell of Acetobacter tropicalis (20% *w/w* biomass suspension with 1% S/L ratio of BR solution)(5000×). Table S1. EDX data for chemical composition (mass%) present in BR before and after bioleaching.

**Author Contributions:** Conceptualization, K.K., T.L., L.-A.T., and M.O.-P.; methodology, T.L. and K.K.; validation, L.-A.T., M.O.-P.; measurements and analysis, K.K., T.L., C.P., and K.P.; writing—original draft preparation, K.K., T.L., and L.-A.T. writing—review and editing, L.-A.T., K.-M.O., M.O.-P., and G.L.; supervision, M.O.-P. All authors have read and agreed to the published version of the manuscript.

**Funding:** This research was partially funded by the European Community's Horizon 2020 Program (H2020/2014-2020), under Grant Agreement number 730105, EC/EASME) (SCALE project).

**Institutional Review Board Statement:** Not applicable.

**Acknowledgments:** The Biotechnology Laboratory of School of Chemical Engineering (National Technical University of Athens) is gratefully acknowledged for providing the incubator.

**Conflicts of Interest:** The authors declare no conflict of interest.

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
