# Peer review of "Study of Microbial Cultures for the Bioleaching of Scandium from Alumina Industry By-Products"

_metals, doi:10.3390/met11060951_

Round 1

Reviewer 1 Report

The authors studied the recovery of critical metals from industrial bauxite residue resulting from the Bayer process. The investigated proccess includes a bioleaching operation using Acetobacter Tropicalis, the results showed around 40 % of scandium dissolution and the production of acetic and citric acid. The research was well planned and could be of interest for the readers of the Journal. Below a list of suggestion to improve the quality of the paper:

  1. The composition of BR is presented in a previous work of the authors. Which batch of BR was used for the present paper? In my opinion a best characterization of BR is useful to better understand the proposed treatment. Among REEs only scandium was detected? 
  2. What could be the economic and environmental benefits of this research? Due to high time of bioleaching, is the scalability of the process compatible with the BR production? I suggest adding these observations in the discussion.
  3. Have the authors already performed or are they planning to carry out some experimentations for the recovery of scandium from the bioleaching solution? Add the operations to investigate.
  4. Can organic acids be recirculated in order to increase their concentrations? Could it be suitable for the growth of bacteria? 

Reviewer 2 Report

This article is devoted to an interesting and relevant topic of bio-leaching of REE from Bauxite residue. In general, the work is well written and can be recommended for publication after removing the following comments:

35 not only Bayer process but also sintering

37 As a result of 1 kg or 0.1 kg per ton?

200-201 This sentence is worth rewriting.

250 About 35% of the References are self-citation, which indicates inappropriate self-citation in this paper and a poor review of the literature in the introduction.

Reviewer 3 Report

Manuscript ID: metals-1240075

Title: Study of microbial cultures for bioleaching of scandium from alumina industry by-product

Authors: Kyriaki Kiskira et al.

Title must be changed to “Study of scandium extraction from red mud by Acetobacter Tropicalis bioleaching”

Introduction.

Authors used “the bauxite residue (BR)”, why not red mud (RM). Red mud is the common name for waste from the Bayer process for alumina production. It is necessary to make a replacement of BR to RM throughout the text of the article.

Line 39. Authors must separate references. Not more 2 references for one sentence.

Line 44. Describe in more detail which methods were used, the main parameters and the degree of scandium extraction.

Line 46. Change link for each specific acid to make it easier for readers to find the right reference.

Line 47-57. Avoid a lot of self-citation. No need to reference all of previous research. Use only the most important in your opinion.

Line 55. Balomenos E. et al. have a huge number of good articles on the use of ionic liquids for scandium extraction. No need to link to an unknown conference. Use articles from this group in well-known journals (e.g. Hydrometallurgy).

Line 63-69. Again, a huge amount of self-citation.

Line 71-73. Use links for each type of bacteria.

Materials and Methods

Line 88. What is the “ferroalumina”?

Line 89. Chemical and mineral composition of red mud must be presented. This is subject of research; authors cannot simply link to previous research.

Section 2.3. Write the equation according to which the scandium extraction degree was calculated.

In Table 1 and 2 use 1, 5, 10 and 30 days, however in Figures 1 and 2 use 20 days. Why authors used different duration?

Figure 1. Improve this figure. Don’t use excel software and spline type for curves. Where are the points in figure 1a? There are no errors bar.

Figure 2. The authors added iron extraction, does it Fe(II) or Fe(III)? What about other metals? Al Si, alkali metals (Na, Ca, K)?

Main conclusions:

This article is devoted to the important problem of red mud utilization with the Sc extraction. However, the article does not contain any physical methods of analysis. There are no data on XRD and chemical composition of red mud before and after bioleaching. No SEM images of red mud samples. The article does not contain the composition of the obtained solutions for macro and microelements. There is no discussion on the further extraction of scandium from the obtained solutions. The authors should add a technological flowsheet of the process indicating the optimal condition and indicators of extraction degree of Sc, Fe and other elements and yield of red mud.

Technical errors:

Article is not written in Metals style. It is necessary to improve the text, tables, and figures. Use https://www.mdpi.com/journal/metals/instructions

Round 2

Reviewer 3 Report

The authors answered in detail all the questions. However, I would draw the attention of the authors to the fact that the authors presented most part of the characteristics of the raw red mud in the supplimentary files. This is strange, since the article is very small (only 11 pages), it would be possible to add this information to the article, because no one will read the attached file. But this is the choice of the authors, so be it.

Good luck with future research, I will be happy to read new approaches to red mud utilization.

The article can be accepted into Metals.